# Unraveling the impact of AXIN1 mutations on HCC development: Insights from CRISPR/Cas9 repaired AXIN1-mutant liver cancer cell lines

Ruyi Zhang[1,2], Shanshan Li[1], Kelly Schippers[1], Boaz Eimers[1], Jiahui Niu[1], Bastian V. H. Hornung[3], Mirjam C. G. N. van den Hout[3], Wilfred F. J. van Ijcken[3], Maikel P. Peppelenbosch[1], Ron Smits[1] *

1 Department of Gastroenterology and Hepatology, Erasmus Medical Center Cancer Institute, University Medical Center, Rotterdam, The Netherlands, 2 Yunnan Key Laboratory of Chiral Functional Substance Research and Application, School of Chemistry & Environment, Yunnan Minzu University, Kunming, China, 3 Erasmus Center for Biomics, Erasmus University Medical Center, Rotterdam, The Netherlands

* m.j.m.smits@erasmusmc.nl

**Data Availability Statement:** Original immunoblot images are included in the manuscript as S1 raw images. Original qPCR data, Luciferase results,

## Abstract

### Background

Hepatocellular carcinoma (HCC) is a highly aggressive liver cancer with significant morbidity and mortality rates. *AXIN1* is one of the top-mutated genes in HCC, but the mechanism by which AXIN1 mutations contribute to HCC development remains unclear.

### Methods

In this study, we utilized CRISPR/Cas9 genome editing to repair AXIN1-truncated mutations in five HCC cell lines.

### Results

For each cell line we successfully obtained 2–4 correctly repaired clones, which all show reduced β-catenin signaling accompanied with reduced cell viability and colony formation. Although exposure of repaired clones to Wnt3A-conditioned medium restored β-catenin signaling, it did not or only partially recover their growth characteristics, indicating the involvement of additional mechanisms. Through RNA-sequencing analysis, we explored the gene expression patterns associated with repaired AXIN1 clones. Except for some highly-responsive β-catenin target genes, no consistent alteration in gene/pathway expression was observed. This observation also applies to the Notch and YAP/TAZ-Hippo signaling pathways, which have been associated with AXIN1-mutant HCCs previously. The AXIN1-repaired clones also cannot confirm a recent observation that AXIN1 is directly linked to YAP/TAZ protein stability and signaling.

### Conclusions

Our study provides insights into the effects of repairing AXIN1 mutations on β-catenin signaling, cell viability, and colony formation in HCC cell lines. However, further investigations

colony formation numbers and MTT results, are included in the manuscript, respectively, as S2 original qPCR, S3 original Luciferase, S4 original colony numbers, S5 original MTT values. The RNA sequencing data for this study have been deposited in the European Nucleotide Archive (ENA) at EMBL-EBI under accession number PRJEB64131 (https://www.ebi.ac.uk/ena/browser/view/PRJEB6411).

**Funding:** This research was financially supported by a China Scholarship Council PhD fellowship to Ruyi Zhang (File NO. 201808530490), Shanshan Li (File NO. 201909370083) and Jiahui Niu (File NO. 202007660001).

**Competing interests:** The authors have declared that no competing interests exist.

**Abbreviations:** HCC, Hepatocellular carcinoma; HBV, hepatitis B virus; HCV, hepatitis C virus; SgRNA, single-guide RNAs; HDR, homology-directed repair; DTT, dithiothreitol; ECL, enhanced chemiluminescence; WRE, Wnt Responsive Element; DMSO, Dimethyl sulfoxide.

are necessary to understand the complex mechanisms underlying HCC development associated with AXIN1 mutations.

## Introduction

Hepatocellular carcinoma (HCC) is the most common type of primary liver cancer, accounting for 75%-85% of all cases. It ranks as the fourth most common cause of death from malignant tumors, with higher mortality rates than gastric and esophageal cancer [1, 2]. As of 2018, reports show approximately 841,000 new cases and 782,000 deaths from HCC worldwide annually [1]. Underlying risk factors for HCC are chronic infections of hepatitis B virus (HBV) or hepatitis C virus (HCV), contaminated food with aflatoxin, heavy alcohol intake, obesity, smoking, and type 2 diabetes [3, 4].

The Wnt/β-catenin cascade is an important signaling pathway that is closely linked to tumorigenesis [5–7]. AXIN1 plays a crucial role as a core component of this pathway. It facilitates the degradation of β-catenin by assembling related proteins (APC, GSK3β, CK1) into a destruction complex, thereby preventing β-catenin accumulation in the nucleus where it can regulate the expression of genes that support cell growth. Given its importance in the destruction complex, AXIN1 acts as an important tumor suppressor protein [8, 9]. Accordingly, numerous studies have consistently shown that AXIN1 mutations significantly contribute to the development of human cancer, particularly HCC, in which mutations are observed in about 8–10% of cases [8].

However, different research groups have reached opposing conclusions on the mechanisms by which AXIN1 regulates the growth and progression of HCC. At first, given its involvement in the breakdown complex, AXIN1 mutations were considered to contribute to cancer by activating β-catenin signaling. However, AXIN1 mutant liver cancers rarely show a marked nuclear accumulation of β-catenin, often regarded as the hallmark of active signaling [10, 11]. In addition, RNA expression profiling of HCCs revealed that AXIN1-mutated cancers show no or at most a modest increase in β-catenin target gene expression above a set threshold [12]. Instead, this latter manuscript suggested that AXIN1 mutant cancers are dependent on the activation of the Notch and YAP/TAZ signaling pathways rather than the activation of typical β-catenin target genes. Thus, these reports suggest that β-catenin signaling is barely increased following AXIN1 mutation and is not relevant for HCC growth. However, Qiao and coworkers showed that AXIN1-driven HCC development in mice is almost entirely dependent on functional β-catenin [13]. Others argued that this result was only obtained in the context of simultaneous overexpression of the proto-oncogene MET [14, 15]. In further support, we and others have observed increased β-catenin signaling in AXIN1-mutant HCC cell lines, albeit modest compared to β-catenin mutant HCCs [9, 16, 17]. This can be explained by a partial functional compensation by the AXIN2 protein. Accordingly, siRNA mediated *AXIN2* knockdown leads to a dramatic increase in β-catenin signaling in AXIN1-mutant HCC cell lines [16, 17]. As such, AXIN1-mutant cancers are strongly dependent on AXIN2 to counterbalance signals that induce β-catenin signaling, and more vulnerable to aberrantly increase this signaling pathway.

Taken together, despite two decades of research of AXIN1-driven HCC development, it is still unclear if AXIN1-mutant HCCs show increased β-catenin signaling and whether this is relevant to support their growth. Furthermore, if β-catenin signaling is not the main culprit through which AXIN1 mutation affects HCC growth, then which other mechanisms may be involved. Previously, we have repaired a truncating AXIN1 (p.R712*) mutation in the SNU449 liver cancer cell line [17]. AXIN1 repair led to reduced β-catenin signaling, but this was not

associated with clear alterations in growth characteristics. This cell line expresses however a long truncated AXIN1 protein, which is expected to retain some functionality [18]. Therefore, we decided to use the CRISPR/Cas9 genome editing technique to repair five HCC cell lines with truncated AXIN1 mutations at more N-terminal locations. The primary objective was to determine whether AXIN1 mutations support HCC development through the Wnt/β-catenin pathway or potentially through other mechanisms.

## Materials and methods

### Cell culture

HEK293T, JHH7, HuH1, SNU423, and Hep3B cells were cultured in DMEM supplemented with 10% (v/v) fetal bovine serum, while JHH6 cells were cultured in William's E medium supplemented with 10% (v/v) fetal bovine serum and 1% UltraGlutamine. Dr. Sandra Rebouissou, Paris, France, generously provided the JHH6, JHH7, HuH1, and SNU423 cell lines. Cells were cultured in a humidified incubator maintained at 37°C with 5% CO2. All cell lines tested negative for mycoplasma based on the real-time PCR method at Eurofins Genomics (Konstanz, Germany). Identity of all cell lines and clones thereof, was confirmed by the Erasmus Molecular Diagnostics Department, using Powerplex-16 STR genotyping (Promega, Leiden, The Netherlands). The exact nature of the *AXIN1* mutation was determined by genomic DNA PCR using the primers reported in **S1 Table**, followed by conventional Sanger sequencing, results of which are shown in **S1 Fig**.

To prepare conditioned medium, HEK293T-R-spondin, L-Control, and L-Wnt3A cells were cultured in complete DMEM medium, and the medium was collected and filtered following previously described procedures [19].

### Expression plasmids used in the present study

Wild-type FLAG-AXIN1 (cat.#109370) was purchased from Addgene, which was used to generate the D94_Q108del variant as previously described [18]. The EGFP-APC plasmid was constructed previously [20].

### Immunoprecipitation

HEK293T cells in a 6-well plate were transiently transfected with 200 ng of FLAG-AXIN1 variants or empty plasmid control, and equal amounts of GFP-APC. After 48 h, cells were washed by cold PBS once, and then 500 μL of cold lysis buffer (30 mM Tris-HCl, pH 7.4; 150 mM NaCl; 1% Triton-100; 5 mM EDTA; 5 mM NaF) containing Halt Protease and Phosphatase Inhibitor Cocktail (100×, cat.#78442, Thermo Fisher Scientific) was added to each well for 15 min on ice. Cells were collected by scraping, transferred into low-adhesion tubes and lysate was cleared at 4°C, by centrifugation at 11,000 g for 15 min. From the cleared lysate, 10% was taken as input control, to which the same volume of 2×Laemmli/0.1M dithiothreitol (DTT) was directly added, followed by heating for 7 min at 95°C. To the remainder of the supernatant, we added 50 μL prewashed ANTI-FLAG® M2 Affinity Gel (cat.#A2220, Sigma-Aldrich), followed by incubation at 4°C for 2 h. Next, FLAG-beads were centrifuged and washed with lysis buffer for 3 times. Finally, the pellet was dissolved in 75 μL 2×Laemmli sample buffer with 0.1M DTT and heated.

### CRISPR/Cas9 mediated repair of AXIN1 mutation in HCC cell lines

CRISPR/Cas9 genome editing was performed as described previously [17]. Briefly, single-guide RNAs (sgRNAs) were designed in the vicinity of *AXIN1* mutations, using the following

CRISPR design tool (http://crispor.tefor.net/), and cloned into pSpCas9(BB)-2A-Puro (PX459, Addgene, cat. #62988). sgRNAs Primers sequences are presented in **S2 Table**. For homology-directed repair (HDR), a 2.1, 1.34 and 0.77 kb genomic PCR fragment, respectively, encompassing exons 2, 3 and 4 of *AXIN1*, was cloned into the pGEM-T Easy Vector (cat.#A1360, Promega). Primers sequences to clone these exons are present in **S1 Table**. Modifications that introduced silent amino acid or PAM-site mutations (**S3 Table**), were generated using Q5 site-directed mutagenesis (NEB).

Cell lines were seeded in 3 wells of a 6-well plate at 40% confluency. Once the cells reached 80% confluency, we transfected them with 1 μg of PX459 plasmid and 5 μg of the HDR plasmid using a 3:1 ratio of Lipofectamine 2000 reagent. After 6 h, the cells were trypsinized and transferred equally into 9 dishes (x200 Petri dishes 100 X 20 mm style Falcon, cat. #353003, Fisher Scientific) at cell densities of 1/7, 2/7, and 4/7, respectively. Puromycin was added to the cells 48 h later to select 2 days for transfected cells. Cells were constantly cultured containing 1:10 diluted L-Wnt3A and R-spondin conditioned medium to maintain high levels of β-catenin signaling. After 2–3 weeks, DNA from clones grown successfully was isolated using the QuickExtract™ DNA Extraction Solution (Epicentre). Once successfully repaired clones were identified, Wnt3A and R-spondin were removed from the medium and clones were further maintained in basal medium. Generation of the JHH7 AXIN1-repaired clones has also been described previously [18]. **S1 Fig** and **S4** and **S5 Tables** contain HCC cell line information and images, concentration of puromycin used, *AXIN1* mutation type, and the screening primers used to identify correctly modified clones both on DNA and cDNA levels. HCC cell line information and images are obtained from the Zucman lab website (https://lccl.zucmanlab.com/hcc/cellLines).

## Western blotting

Cells were lysed in 2× Laemmli Sample Buffer (4% SDS, 20% glycerol, 0.004% bromophenol blue, 0.15M Tris-Cl, pH 6.8) with 0.1 mol/L DTT and heated 7 min at 95˚C. Protein samples were run in 10% sodium dodecyl sulfate-polyacrylamide gel electrophoresis (SDS-PAGE), and transferred onto Immobilon-PVDF membranes (Millipore). For AXIN1 western blot analysis we used an enhanced chemiluminescence (ECL)-based detection method. Membranes for ECL detection were blocked and incubated using Immobilon Block-CH reagent (cat. #WBAVDCH01, Millipore). The primary antibody was incubated at 4˚C overnight. Next, membranes were washed with PBS containing 0.05% Tween 20 (PBST) for 10 min, 3 times. The secondary antibodies were Goat Anti-Rabbit Immunoglobulins/HRP (1:10,000, cat. #P044801-2, Agilent Technologies Netherlands BV) or Goat Anti-Mouse Immunoglobulins/ HRP (1:10,000, cat.#P026002-2, Agilent Technologies Netherlands BV). The membranes were washed twice with 0.05% PBST for 15 min and once with PBS for 10 min. Membranes were then incubated with Immobilon ECL Ultra Western HRP Substrate (cat.#WBULS0100, Millipore) and visualized using an Amersham Imager 600 (GE Healthcare).

For fluorescent western blotting, membranes were blocked with Odyssey blocking buffer (cat.#927–70001, Licor-Biosciences). Secondary antibodies used were IRDye 680 Goat anti-Mouse (1:10,000, cat.#926–68070, Licor-Biosciences) or IRDye 800 Goat anti-Rabbit (1:10,000, cat.#926–32211, Licor-Biosciences).

The primary antibodies used in this study were anti-β-actin (1:1000, cat.#sc-47778, Santa Cruz Biotechnology), anti-α-Tubulin (11H10) Rabbit mAb (1:1000, cat.#2125S, Cell Signaling Technology), anti-AXIN1 (1:1000, cat.#2087, Cell Signaling Technology), anti-FLAG (1:1000, cat.#F1804, Sigma-Aldrich), anti-GFP (D5.1) Rabbit mAb (1:1000, cat.#2956S, Cell Signaling Technology), Hippo Signaling Antibody Sampler Kit (1:1000, cat.#8579, Cell Signaling

Technology). Proprietary epitope of the AXIN1 antibody (cat.#2087) is between Q630 and E760, based on information provided by Cell Signaling Technology.

## β-catenin reporter assays

We conducted the β-catenin luciferase reporter assays according to previously reported methods [17]. Briefly, when the cells reached a confluence of 70–80%, we co-transfected them with 100 ng of Wnt Responsive Element (WRE) vector or Mutant Responsive Element (MRE) and 10 ng of CMV-Renilla plasmid in a 24-well plate using Lipofectamine 2000 reagent (cat. #10696153, Thermo Fisher Scientific). After 48 h, we measured the β-catenin reporter activity using the Dual-Luciferase® Reporter Assay System (cat.#E1910, Promega), following the manufacturer's instructions. The β-catenin reporter activities are presented as WRE/CMV-Renilla or WRE/MRE ratios. However, for Hep3B cells, we tested them in a 12-well plate and transfected them with 300 ng of WRE vector or MRE and 30 ng of CMV-Renilla plasmid. To perform siRNA-mediated *AXIN2* knockdown, we used a final concentration of 20 nM of siRNA per well in a 24-well plate; ON-TARGET plus Non-targeting Pool (cat.#D-001810-10-05), ON-TARGET plus SMART pool human AXIN2 siRNA (cat.#L-008809-00-0005).

## Quantitative real-time PCR (qRT-PCR)

RNA isolation and q-RT-PCR were basically performed as previously described [21]. RNA was extracted using the NucleoSpin® RNA isolation kit from Macherey-Nagel (Dueren, Germany) and transcribed into cDNA using the PrimeScript™ RT Master Mix (Perfect Real Time, Takara, cat.#RR036A) according to the manufacturer's instructions. Quantitative real-time PCR (qRT-PCR) was performed on a StepOnePlus™ Real-Time PCR System (Applied Biosystems) using SYBR™ Green PCR Master Mix (ThermoFisher). The gene expression was calculated using the $2^{-\Delta\Delta T}$ method with means of technical replicates. *GAPDH* was used as a reference gene for experimental models. The primer sequences used for qPCR are listed in **S6 Table**.

## MTT assay

To determine the baseline cell viability, we seeded HCC cells at a concentration of 1000 cells per well in 100 μL of medium in a 96-well plate. After 1, 3, and 7 days, we tested the cell viability by adding 5 mg/mL of MTT (Thiazolyl blue tetrazolium bromide, cat.#M5655, Sigma) and incubating the cells at 37˚C for 3 h. We removed the medium and added 100 μL of Dimethyl sulfoxide (DMSO), followed by shaking for 10 min. We used a microplate absorbance reader (Bio-Rad, Hercules, CA, USA) to determine the absorbance at a wavelength of 490 nm.

To assess the impact of L-Wnt3A cell growth, we tested the effect of 10% L-Wnt3A conditioned medium (CM), using L-Control conditional medium as a control.

## Colony formation

After trypsinizing the cells, we seeded them at a concentration of 1000 cells per well in a 6-well plate. We refreshed the culture medium every 3 days until we could observe the growth of visible colonies, which took approximately two weeks. After that, we washed the cells with PBS, fixed them, and stained them with a solution containing 0.1% crystal violet in PBS with 20% methanol. After 30 min, we washed the stained plate with ddH$_2$O and air-dried it for another 30 min. Finally, we used Gelcount (Oxford Optronix Ltd.) to automatically count the colonies as previously described [17].

## RNA sequencing

RNA was isolated by the NucleoSpin® RNA isolation kit of Macherey-Nagel (Dueren, Germany), according to the manufacturer's instructions. At least 2 μg of RNA samples were prepared and service sequenced at Macrogen Europe BV, Amsterdam, the Netherlands. Prior to library preparation using the TruSeq Stranded mRNA kit, the RNA underwent a sample quality check using Agilent TapeStation 2200 to evaluate its RNA Integrity Number and total quantity. The prepared libraries were further subjected to a Library Quality Control using Agilent Technologies 2100 Bioanalyzer and a DNA 1000 chip to assess the library size and quantified using qPCR according to the Illumina qPCR Quantification Protocol Guide. Finally, the libraries were sequenced on a NovaSeq6000 S4 with parameters set at 100bp PE and 4Gb of throughput per sample. Illumina adapters and poly-A sequences were trimmed off the reads with the in-house tool AdapterTrimmer, if at least 2 bases from the end of a read were matching, with a maximum of 2 mismatches over the whole matched region. Reads were discarded if they were trimmed to 19 bases or shorter. Afterwards reads were mapped against the GRCh38 human reference using HiSat2 (version 2.2.1) [22]. Gene expression values were called using htseq-count (version 0.12.4) [23], with reads counted only on the reverse strand, and in union mode. Ensembl release 101 was used as the annotation. Samtools v1.11 was used throughout the workflow to sort reads, to convert file formats and to obtain statistics [24]. Further statistics were obtained with the R environment for statistical computing [25], version 4.2.1, using the packages tidyverse [https://github.com/tidyverse] version 2.0.0 and stringr 1.5 [https://github.com/tidyverse/stringr]. Differential expression analysis was performed with DESeq2, version 1.36 with vst normalization [26], and a q-value cutoff of 0.01 and a log2 fold-change cutoff of at least 1 was applied. Python3 with the matplotlib package version 3.5.1 was used for data analysis [27].

## Statistical analysis

Statistical analyses were carried out using software GraphPad Prism version 8.0.2 (GraphPad Software Inc., San Diego, California, USA). In this study, the expression of continuous variables was presented as the mean ± standard deviation (SD). Comparisons between groups were performed with the Mann-Whitney test. Differences were considered significant at a P value less than 0.05 ($*P \leq 0.05$, $**P \leq 0.01$, $***P \leq 0.001$, $****P \leq 0.0001$).

## Results

### Baseline characteristics of employed AXIN1-mutant HCC cell lines

To investigate the role of AXIN1 mutation in HCC tumorigenesis, we selected five cell lines with homozygous *AXIN1* mutations (**Fig 1A/1B and S1 Fig**). All mutations were confirmed to be homozygous, most likely resulting from mitotic recombination during tumorigenesis. As previously reported by Caruso et al. [28], these five HCC cell lines can be divided into two groups: JHH7, Hep3B, and HuH1 belong to the hepatoblast-like group, while JHH6 and SNU423 are mesenchymal-like. JHH6, Hep3B, HuH1, and SNU423 all carry mutations predicted to lead to short truncated proteins lacking most functional AXIN1 domains, while JHH7 carries a homozygous D94_Q108del AXIN1 deletion within the APC binding domain (**Fig 1A/1B and S1 Fig**). Accordingly, western blot analysis using an AXIN1 C-terminal antibody revealed that four out of five HCC cell lines did not express wild-type AXIN1, while JHH7 shows an AXIN1 band (**Fig 1C**). To confirm the defective nature of this latter mutant protein, we previously showed that a construct expressing this variant leads to loss of APC binding and a clear increase of β-catenin reporter activity (**S2 Fig**) [18]. **Fig 1D** shows the

**A**

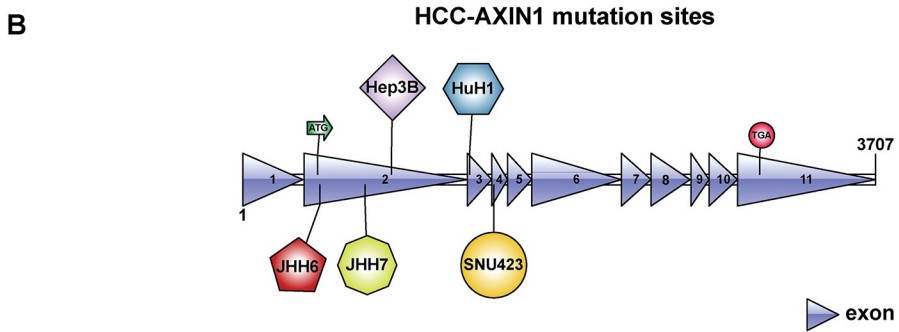

| Cell Line | Accession | Gene | AA Mutaion | CDS Mutation | Zygosity | Type | Exon location | Differentiation |
|-----------|-----------|------|------------|--------------|----------|------|---------------|-----------------|
| JHH6 | CVCL_2788 | *AXIN1* | p.E5D+Q6* | c.16C>T | Homozygous | Substitution - Nonsense | exon 2 | Mesenchymal-like |
| JHH7 | CVCL_2805 | *AXIN1* | p.D94_Q108del | c.282_326del | Homozygous | Deletion - In frame | exon 2 | Hepatoblast-like |
| Hep3B | CVCL_0326 | *AXIN1* | p.R146* | c.436C>T | Homozygous | Substitution - Nonsense | exon 2 | Hepatoblast-like |
| HuH1 | CVCL_2956 | *AXIN1* | p.R298Lfs*112 | c.893_906delinsT | Homozygous | Deletion - Frameshift | exon 3 | Hepatoblast-like |
| SNU423 | CVCL_0366 | *AXIN1* | p.P345Vfs*65 | c.1033_1045del | Homozygous | Deletion - Frameshift | exon 4 | Mesenchymal-like |

**B**

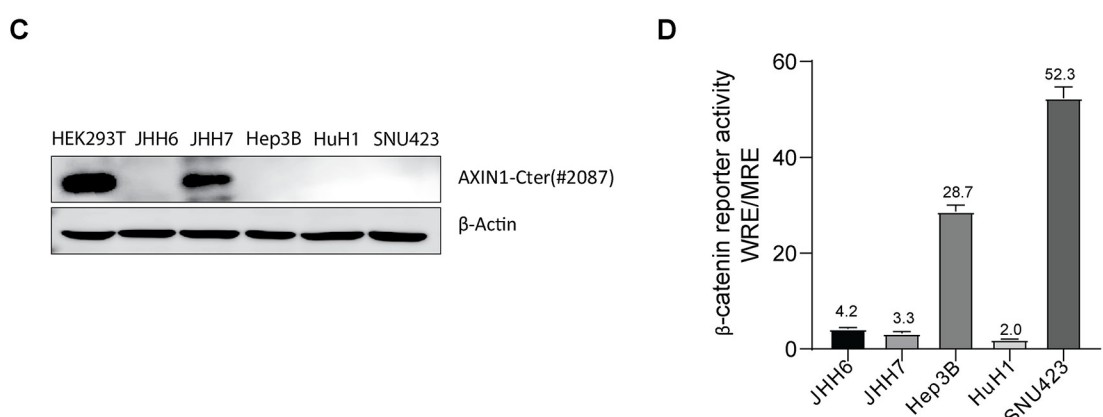

**C**

**D**

Fig 1. Baseline characteristics of AXIN1-mutant HCC cell lines used in this study. (A) Information about the type and zygosity of the AXIN1 mutation present in five employed HCC cell lines. Mesenchymal-like and hepatoblast-like differentiation is according to the paper by Caruso et al. [28]. (B) Diagram depicting the location of AXIN1 mutations in each HCC cell line. (C) Western blotting using a C-terminal AXIN1 antibody reveals the absence of AXIN1 in four cell lines and expression of a mutant AXIN1 in JHH7. HEK293T serves as the control for wild-type AXIN1 protein expression. β-Actin was used as a loading control. (D) A β-catenin reporter assay was conducted in HCC cells to determine the baseline levels of β-catenin signaling. Reporter activities are represented as WRE/MRE ratios (n = 3, three independent experiments, mean ± SD).

baseline β-catenin reporter activity in these HCC cell lines, indicating that all lines have some evidence of nuclear signaling.

## Repairing AXIN1 mutations results in reduced β-catenin signaling

Using CRISPR/Cas9 gene editing, we successfully obtained 2–4 independent clones with repaired *AXIN1* mutation for all five HCC cell lines. Most clones were homozygously repaired with some exceptions (S3 Fig). In HuH1 clone 2B3, two out of three chromosomes were correctly repaired, while cDNA analysis only revealed expression of the repaired transcript. For the SNU423 cell line, we achieved a heterozygous repair. Immunoblot analysis confirmed that the AXIN1-repaired clones show restored AXIN1 protein expression (Fig 2A). To determine the effect on β-catenin signaling following AXIN1 repair, we used quantitative real-time PCR

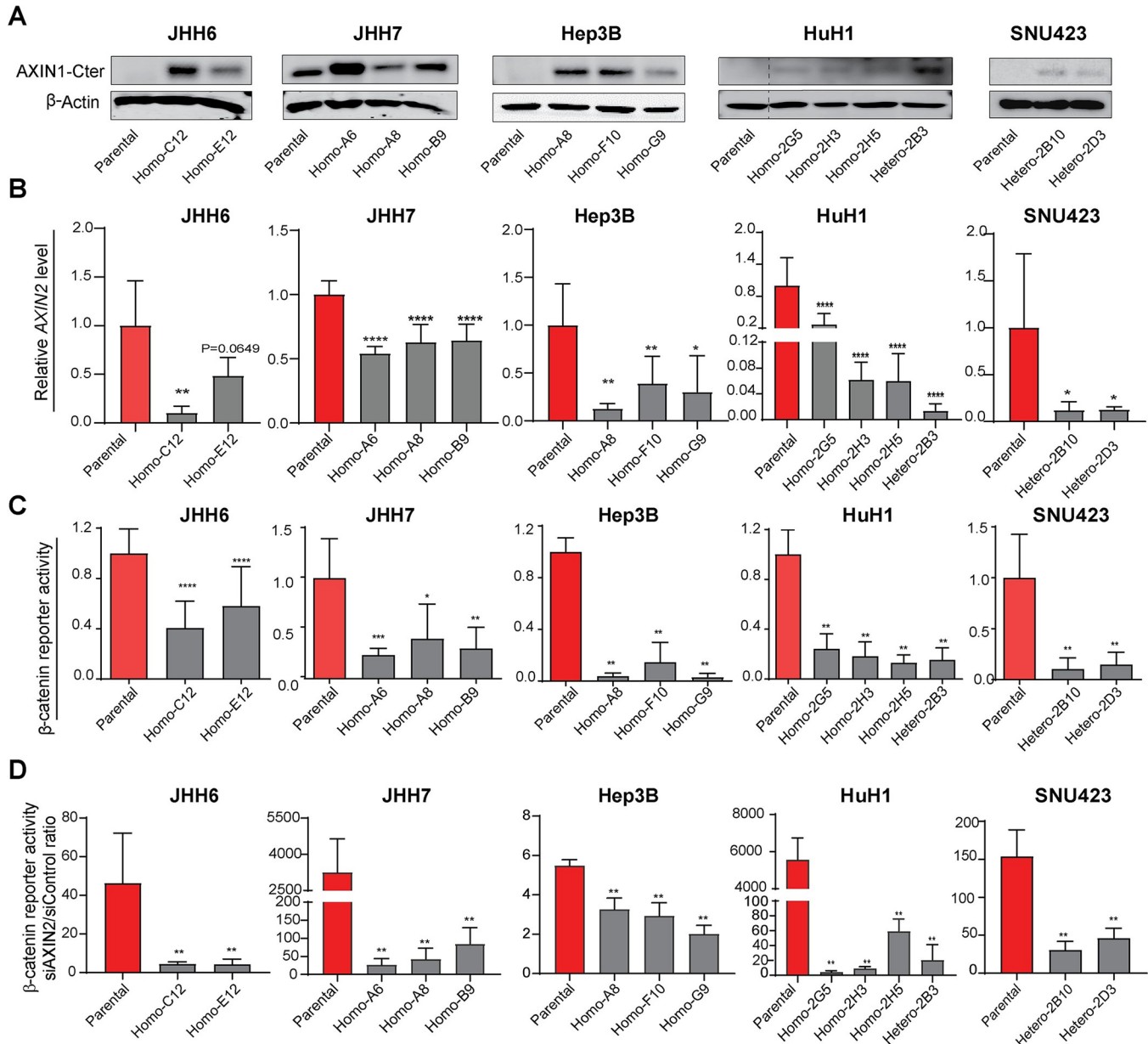

**Fig 2. HCC cell lines with repaired AXIN1 mutation show reduced β-catenin signaling.** (A) Western blot analysis revealing restored expression of endogenous wild-type AXIN1 in Crispr-Cas9 AXIN1-repaired HCC cell lines. (B) qPCR was used to detect the difference in *AXIN2* mRNA expression level between parental and repaired clones (mean ± SD, n = 3, three independent experiments). (C) A β-catenin reporter assay shows a significant decrease in β-catenin signaling in all AXIN1-repaired clones. The WRE/CMV-Renilla ratio for the parental AXIN1-mutant clones was arbitrarily set to 1 for each cell line, to which all β-catenin reporter WRE/CMV-Renilla ratios for repaired clones were normalized (mean ± SD, n = 3, three independent experiments). (D) To determine the dependence on AXIN2 to counterbalance β-catenin signaling in all clones, we performed siRNA-mediated knockdown of *AXIN2*. The parental AXIN1-mutant lines show a robust and much higher activation of β-catenin signaling than their corresponding repaired clones, indicating that they are much more sensitive to activate β-catenin signaling. WRE/CMV-Renilla ratios were obtained for siAXIN2 and siControl for each clone. Next, for each clone this WRE/CMV ratio was set to 1 for the siControl value, to which the siAXIN2 value was normalized. Finally, the figure shows the normalized siAXIN2/siControl ratio (mean ± SD, n = 6, two independent experiments). Statistical significance for all experiments was analyzed using a Mann-Whitney test (*$P < 0.05$, **$P < 0.01$, ***$P < 0.001$, ****$P < 0.0001$).

(qRT-PCR) to assess the expression levels of *AXIN2*, a well-known target gene of β-catenin. As depicted in **Fig 2B**, expression of *AXIN2* mRNA was significantly reduced in all AXIN1-repaired clones, except for JHH6 clone E12. Furthermore, the expression of *NOTUM*, a key target gene of the β-catenin pathway elevated in HCC cells [29], was also significantly reduced in all AXIN1-repaired samples, again with the exception of JHH6 clone E12 (**S4 Fig**). Finally, our findings were further reinforced by a marked decrease in β-catenin reporter activity (**Fig 2C**). For JHH7, the immunoblot, *AXIN2* qPCR and β-catenin reporter assay have also been shown by us previously, but are reproduced here for clarity [18]. Taken together, these results show that AXIN1 repair in all five cell lines leads to a significantly reduced level of β-catenin signaling.

Next, we performed a siRNA-mediated knockdown experiment of *AXIN2* in all parental and AXIN1 repaired clones. AXIN2 can partially compensate for the functional loss of AXIN1 [16, 17]. Accordingly, the fold change in β-catenin reporter activity is much more pronounced in the original AXIN1 mutant cells than in the repaired clones (**Fig 2D and S5 Fig**). This confirms that AXIN1 mutant HCC cells are strongly dependent on AXIN2 to counterbalance signals that induce β-catenin signaling, and are more vulnerable to aberrantly increase this signaling pathway.

## AXIN1-repaired clones grow slower than their mutant counterparts

Several studies have shown that AXIN1 mutation contributes to the growth and progression of HCC [9, 11–13]. To verify whether this also holds true for the AXIN1 mutant cell lines, we assessed cell viability at 1, 3, and 7 days using a MTT assay (**Fig 3A**). The data revealed that the parental cells experienced a notable boost in cell growth at the 7-day time point compared to the AXIN1-repaired clones. Similar findings were obtained in a colony formation assay. In all cases, the original mutant cells formed more colonies, which mostly were also of larger size

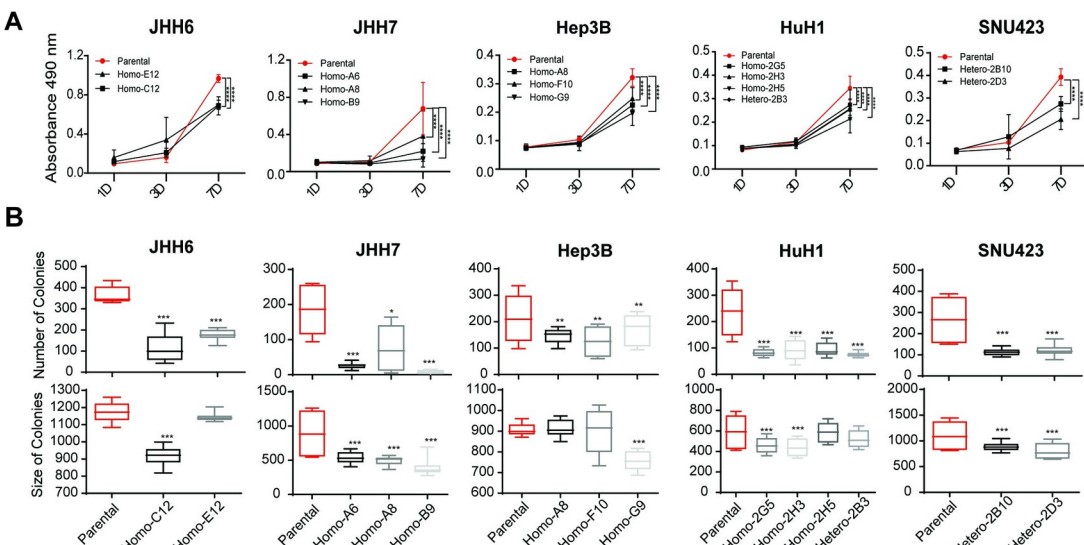

**Fig 3. AXIN1-repaired clones grow slower than original mutant ones.** (A) Cell growth in the parental and repaired clones was measured using a MTT assay after 1, 3, and 7 days (mean ± SD, n = 6, three independent experiments). Statistical significance for 7-day timepoint was analyzed using a Mann-Whitney test (****$P < 0.0001$). (B) A colony formation assay shows significantly reduced colony numbers for all AXIN1-repaired clones. Also, the average colony size was reduced for most clones (mean ± SD, n = 6, two independent experiments). Statistical significance for all experiments was analyzed using a Mann-Whitney test (*$P < 0.05$, **$P < 0.01$, ***$P < 0.001$).

(**Fig 3B** and **S6 Fig**). Thus, these findings demonstrate that AXIN1 mutation confers a clear growth advantage onto liver cancer cells, which correlates with increased β-catenin signaling.

## Exposing AXIN1 repaired cells to Wnt3A does not or only partially restore their growth

As our findings suggest that AXIN1 mutation plays a role in HCC by affecting the Wnt/β-catenin signaling pathway, we hypothesized that increasing β-catenin signaling would restore the growth characteristics of AXIN1-repaired clones to their original AXIN1-mutant levels. To this aim, we made use of L-Wnt3A conditioned medium, which was able to restore β-catenin signaling in repaired clones to levels comparable with the original mutant lines (**Fig 4A**). However, cell growth, as measured by MTT and colony formation, did not or only partially recover to parental levels in all repaired clones after Wnt3A treatment (**Figs 4B and 5**). The MTT assay showed that growth of all JHH6 and SNU423 clones was unchanged or even slightly reduced. Colony numbers are also unaffected by Wnt3A for SNU423, while they are significantly increased for both JHH6 repaired clones. Interestingly, the JHH6 parental line shows a significant reduction in colony numbers. For JHH7 we observe a partial recovery in colony numbers for all repaired clones. However, the MTT-assay shows that only clone A6 approaches that of the parental cells, while the other two clones are not clearly altered by Wnt3A. Exposing AXIN1-repaired Hep3B cells to Wnt3A leads to a comparable growth rate as untreated parental cells in the MTT-assay, but does not alter the number of colonies formed. A similar observation is made for 3 of 4 Wnt3A-treated repaired HuH1 clones that grow comparable to the parental cells, while no clear alteration is observed in colony formation.

Thus, we observe a variable response between cell lines in the degree of growth restoration that can be achieved by adding Wnt3a. In none of the cases are both the MTT-assay and colony formation revived to the level of parental cells. Both mesenchymal-like cell lines, that is JHH6 and SNU423, show the least restoration to parental levels following re-activation of β-catenin signaling. Taken together, this implies that reduced β-catenin signaling resulting from AXIN1-repair is at most partially responsible, and suggests that other mechanisms are at play that are more relevant or cooperate with β-catenin signaling to inhibit cell growth in AXIN1-repaired cells.

## RNA sequencing analysis

As reduced β-catenin signaling could not fully explain the altered growth characteristics of AXIN1-repaired cells, we explored other genes or signaling pathways that may be affected by AXIN1. To this aim, we used RNA sequencing combined with a detailed investigation of specific genes/pathways. Principal component analysis of the RNA sequencing data clearly split the samples in five cell line related groups (**S7A Fig**), indicating that the cell line identity affected gene expression more strongly than AXIN1-mutation status. Within each group, the repaired clones show somewhat more variation compared with the parental samples. The number of differentially expressed genes (threshold FDR 0.01, at least a log2 fold change of 1) is depicted in **S7 Table**, and identifies between 68–283 upregulated and 37–273 downregulated genes per cell line (**S7B Fig**). Comparing differentially expressed genes between cell lines shows little consistency in the identified genes. Most genes are altered uniquely in only one cell line, while at most 25 genes are upregulated and 10 being downregulated in at least 2 cell lines at the same time (**S7 Table**). Likewise, a KEGG analysis does not reveal a pathway consistently altered in all cell lines (**S8 Fig**). Thus, the RNA sequencing analysis does not indicate a specific set of genes/pathways that are clearly affected by AXIN1 mutation. Below we look into more detail to genes/pathways that have been linked to AXIN1 or AXIN1-mutant HCCs.

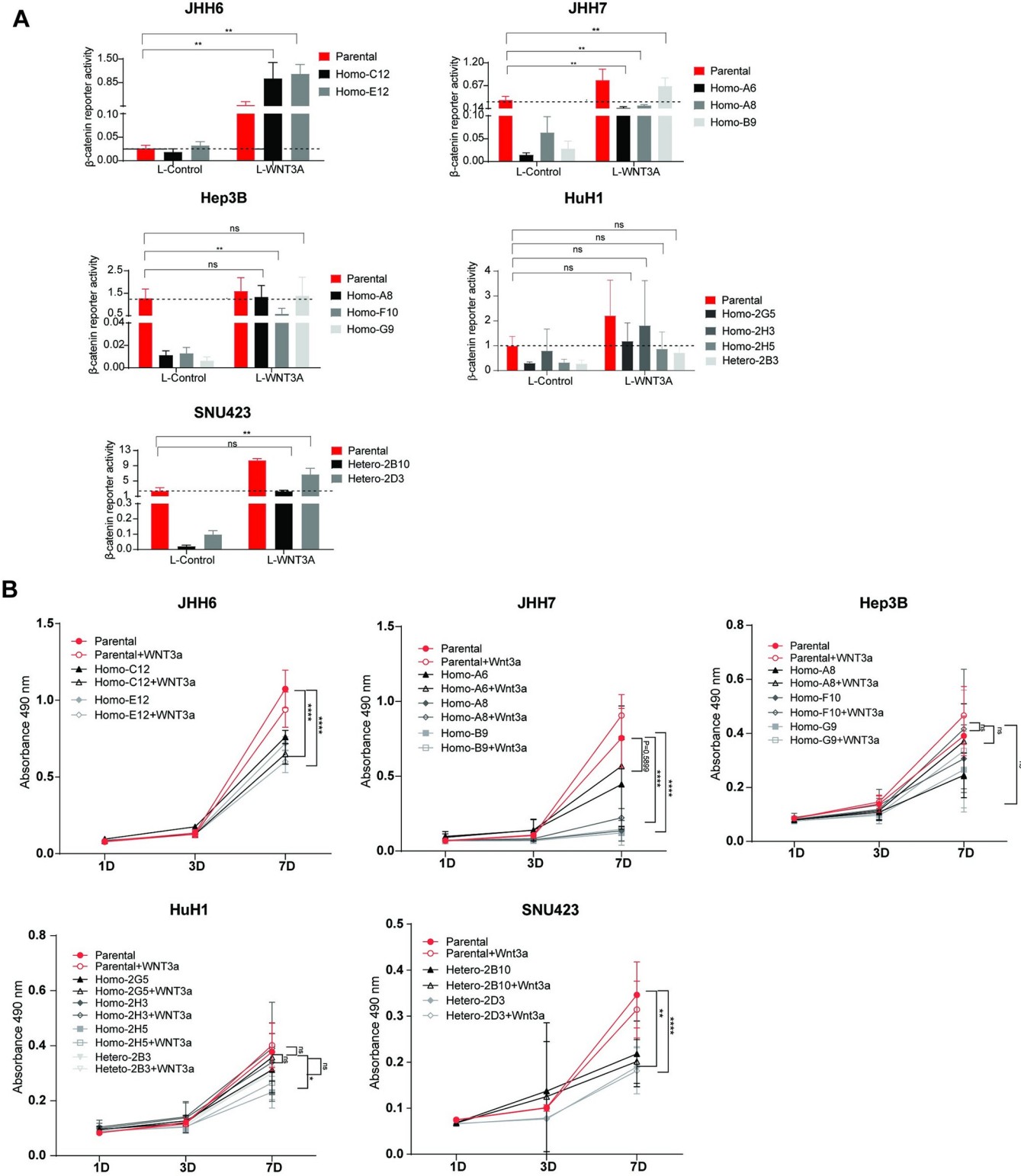

**Fig 4. Adding Wnt3A can restore β-catenin signaling in AXIN1 repaired cell clones, but not or only partially revive the growth of AXIN1 repaired cell clones.** A β-catenin reporter assay was conducted to evaluate the impact of L-Wnt3a treatment on β-catenin signaling in HCC cells. The results are presented as the ratio of WRE/CMV-Renilla (mean ± SD, n = 3, two independent experiments). (B) An MTT assay was performed to measure cell growth in the AXIN1-repaired clones (mean ± SD, n = 6, two independent experiments). A statistical analysis comparing the parental cells without Wnt3A to the repaired clones with Wnt3A was carried out using a Mann-Whitney test (*$P < 0.05$, **$P < 0.01$, ****$P < 0.0001$).

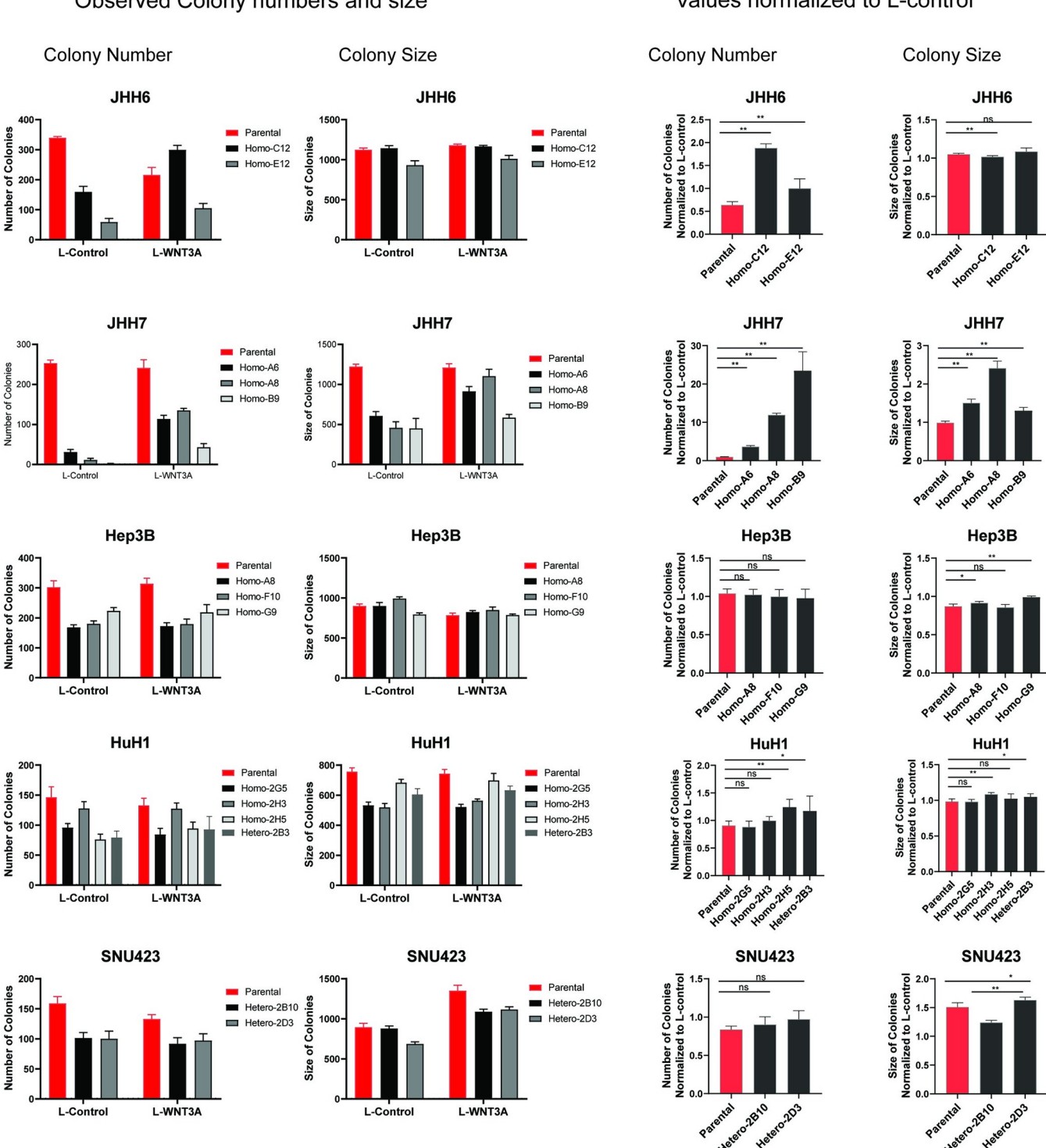

**Fig 5. Colony formation was assessed in all cell lines and clones thereof with and without L-Wnt3A conditioned medium.** The results on the left panel display the observed colony numbers and size, while the right panel shows normalized values relative to L-control (n = 3, two independent experiments). Statistical significance for all experiments was analyzed using a Mann-Whitney test (*$P<0.05$, **$P<0.01$).

## Wnt/β-catenin signaling

Previously, a 23-gene signature was used to explore β-catenin signaling in liver cancers, including canonical, liver-specific and negatively regulated β-catenin target genes [12]. **Fig 6A** shows 17 of these genes identifiable in our expression analysis, in which genes downregulated in the AXIN1-repaired clones are marked in red, while upregulated ones are in green. In accordance with our qPCR analysis, all clones, except one JHH6 clone, show reduced expression of *AXIN2*. The same holds true for *NKD1*, while *LGR5* is clearly reduced in all AXIN1-repaired JHH7 and Hep3B clones, undetectable in HuH1 and JHH6, and surprisingly increased in SNU423. With the exception of Hep3B, in which all canonical β-catenin target genes are reduced in expression, all other cell lines show a variable response for the remaining canonical target genes. The latter is also the case for the liver-specific β-catenin target genes and somewhat for the negatively regulated genes *HAL* and *GLS2*.

**Hippo-YAP/TAZ signaling.** Hippo-YAP/TAZ signaling is an important pathway to control organ size and tissue growth. In a phosphorylation cascade the kinases MST1/2 and LATS1/2, ultimately lead to phosphorylation and breakdown of YAP/TAZ protein, thereby preventing their nuclear signaling activity [30]. Previous research reported that most AXIN1-mutant HCCs have a gene signature enriched in the YAP/TAZ pathway with a strong increase of target gene expression [12]. Moreover, YAP/TAZ protein stability was suggested to be directly regulated by AXIN1 through association with the C-terminus of AXIN1 [31, 32]. Surprisingly, our RNA-seq data do not support a direct association between AXIN1 mutation status and increased YAP/TAZ signaling. Well-established target genes, such as *CCN1* (*CYR61*), *CCN2* (*CTGF*) and *HMMR* are unchanged or even significantly higher expressed in the AXIN1-repaired clones (**Fig 6B, S9 Fig**), which is contrary to expectation. Likewise, immunoblotting shows no alteration in YAP/TAZ levels in JHH6, HuH1 and SNU423 clones, and increased YAP1 levels in all JHH7 clones (**Fig 6C, S10 Fig**). The "expected" decrease in YAP/TAZ levels is only observed in Hep3B AXIN1-repaired clones, which is however accompanied by increased expression of their target genes. Expression levels of Hippo regulatory proteins such as MST1/2, LATS1/2, and MOB1 are also not clearly altered in the AXIN1-repaired clones on RNA or protein levels (**Fig 6B/6C**). Thus, our results indicate that AXIN1 mutation, at least in the investigated cell lines, does neither lead to a consistent increase in YAP/TAZ protein stability nor elevated expression of YAP/TAZ target genes.

## Notch signaling

AXIN1 mutant HCCs were also shown to be enriched for an oncogenic Notch signature [12]. To investigate whether AXIN1 mutation may be directly involved in regulating expression of Notch target genes, we determined their expression in our RNA-seq data (**Fig 7A**). A downward trend was observed for *SPP1* in JHH6, JHH7 and Hep3B AXIN1-repaired clones, and for *HEY1* in JHH7, Hep3B and HuH1. However, for other Notch target genes we do not observe a consistent reduced expression in the AXIN1-repaired samples, which was confirmed by qPCR for *HES1*, *TSPAN8*, and *SPP1* (**Fig 7B**). Taken together, our data do not indicate a direct involvement of AXIN1 mutation in regulating Notch signaling in the investigated cell lines.

## Discussion

The *AXIN1* gene is one of the most frequently mutated genes in hepatocellular cancer [8]. Nevertheless, its exact contribution to tumorigenesis remains uncertain and debated. Inactivating Axin1 in mouse livers leads to late onset formation of a small number of liver cancers [11, 12]. Analysis of these tumors and their AXIN1-mutant human counterparts, has uncovered features more commonly observed in this subset of cancers, such as frequent activation of the

**A**

| GeneName | JHH6 parental | | C12 | E12 | JHH7 parental | | A6 | A8 | B9 | Hep3B parental | | A8 | F10 | G9 | HuH1 parental | | 2B3 | 2G5 | 2H3 | 2H5 | SNU423 parental | | 2B10 | 2D3 |
|---|---|---|---|---|---|---|---|---|---|---|---|---|---|---|---|---|---|---|---|---|---|---|---|---|
| NKD1 | 29 | 23 | 7 | 37 | 248 | 272 | 71 | 57 | 143 | 586 | 553 | 42 | 103 | 26 | 6 | 9 | 0 | 2 | 1 | 0 | 1 | 9 | 3 | 1 |
| AXIN2 | 265 | 267 | 50 | 195 | 245 | 244 | 88 | 98 | 94 | 124 | 93 | 6 | 14 | 15 | 165 | 145 | 10 | 12 | 34 | 10 | 218 | 218 | 65 | 41 |
| LGR5 | 0 | 2 | 2 | 0 | 528 | 517 | 1 | 4 | 28 | 1443 | 1363 | 37 | 63 | 43 | 0 | 3 | 0 | 0 | 0 | 0 | 7 | 3 | 42 | 32 |
| SP5 | 3 | 0 | 5 | 12 | 47 | 39 | 31 | 43 | 33 | 125 | 121 | 21 | 33 | 29 | 79 | 79 | 33 | 67 | 13 | 23 | 11 | 16 | 22 | 19 |
| ZNRF3 | 469 | 477 | 306 | 362 | 1623 | 1601 | 754 | 1283 | 1083 | 1025 | 929 | 447 | 595 | 303 | 287 | 324 | 234 | 192 | 309 | 308 | 256 | 279 | 244 | 234 |
| RNF43 | 2 | 2 | 1 | 0 | 563 | 588 | 361 | 465 | 603 | 969 | 905 | 394 | 439 | 370 | 586 | 658 | 370 | 325 | 766 | 402 | 62 | 54 | 117 | 118 |
| TNFRSF19 | 8 | 4 | 29 | 185 | 142 | 161 | 98 | 330 | 210 | 817 | 875 | 97 | 172 | 192 | 5 | 2 | 4 | 13 | 0 | 195 | 1571 | 1629 | 243 | 332 |
| LEF1 | 0 | 0 | 0 | 0 | 141 | 165 | 252 | 269 | 143 | 133 | 131 | 53 | 40 | 20 | 1 | 1 | 5 | 2 | 0 | 2 | 16 | 15 | 141 | 107 |
| RHBG | 0 | 0 | 0 | 0 | 105 | 98 | 330 | 219 | 270 | 26 | 30 | 56 | 23 | 52 | 27 | 12 | 0 | 6 | 3 | 1 | 3 | 0 | 1 | 0 |
| GLUL | 567 | 582 | 73 | 254 | 7929 | 7796 | 9198 | 5610 | 7672 | 5903 | 5609 | 5064 | 4923 | 5305 | 3685 | 3774 | 3285 | 3653 | 2977 | 2393 | 1566 | 1522 | 2471 | 2396 |
| TBX3 | 942 | 909 | 969 | 831 | 727 | 868 | 658 | 749 | 1021 | 1147 | 1087 | 559 | 601 | 457 | 790 | 773 | 405 | 456 | 332 | 453 | 939 | 929 | 1762 | 1594 |
| LAMA3 | 973 | 953 | 976 | 544 | 8 | 5 | 12 | 3 | 34 | 124 | 133 | 84 | 104 | 239 | 866 | 830 | 501 | 367 | 605 | 1377 | 515 | 481 | 642 | 1074 |
| TRIB2 | 345 | 295 | 172 | 141 | 239 | 289 | 251 | 63 | 349 | 3 | 0 | 2 | 2 | 3 | 0 | 1 | 0 | 0 | 0 | 0 | 33 | 45 | 11 | 28 |
| OAT | 0 | 0 | 0 | 0 | 4936 | 4289 | 7298 | 4841 | 2996 | 353 | 358 | 662 | 552 | 649 | 1002 | 986 | 1007 | 1222 | 1375 | 556 | 4014 | 4341 | 4548 | 4893 |
| CYP2E1 | 0 | 0 | 1 | 1 | 7 | 14 | 23 | 50 | 14 | 6 | 12 | 11 | 5 | 14 | 2 | 1 | 1 | 7 | 1 | 2 | 23 | 18 | 18 | 61 |
| HAL | 2 | 2 | 0 | 5 | 613 | 519 | 1519 | 1155 | 414 | 479 | 445 | 903 | 407 | 940 | 139 | 143 | 261 | 492 | 132 | 192 | 3 | 3 | 2 | 5 |
| GLS2 | 5 | 3 | 8 | 7 | 0 | 0 | 9 | 0 | 1 | 8 | 14 | 10 | 10 | 14 | 6 | 10 | 12 | 9 | 16 | 27 | 18 | 4 | 2 | 7 |

Legend (color coding):
- <0.5
- 0.5–0.66
- 0.66–1.5
- 1.5–2
- \>2

- Canonical β-catenin target genes
- Liver specific **positive** β-catenin targets
- Liver specific **negative** β-catenin targets

**B**

| GeneName | JHH6 parental | | C12 | E12 | JHH7 parental | | A6 | A8 | B9 | Hep3B parental | | A8 | F10 | G9 | HuH1 parental | | 2B3 | 2G5 | 2H3 | 2H5 | SNU423 parental | | 2B10 | 2D3 |
|---|---|---|---|---|---|---|---|---|---|---|---|---|---|---|---|---|---|---|---|---|---|---|---|---|
| CCN1 | 3975 | 5339 | 5813 | 5390 | 203 | 244 | 298 | 1139 | 1160 | 3672 | 3951 | 11060 | 10904 | 11073 | 261 | 241 | 479 | 380 | 200 | 1263 | 5570 | 5324 | 11406 | 6612 |
| HMMR | 3141 | 3156 | 3484 | 2873 | 1382 | 1311 | 1097 | 1687 | 1166 | 499 | 525 | 520 | 767 | 345 | 687 | 727 | 884 | 587 | 910 | 1176 | 1727 | 1803 | 2287 | 2154 |
| CCN2 | 205 | 275 | 249 | 434 | 3212 | 3429 | 7061 | 9018 | 11240 | 4573 | 5281 | 28375 | 23819 | 21965 | 680 | 649 | 1172 | 828 | 422 | 1599 | 2540 | 2546 | 3061 | 3725 |
| AXL | 7441 | 8262 | 7220 | 13940 | 0 | 3 | 20 | 86 | 11 | 37 | 51 | 90 | 58 | 82 | 131 | 129 | 226 | 125 | 313 | 209 | 6254 | 6067 | 8405 | 5906 |
| BIRC5 | 2779 | 2770 | 2872 | 2875 | 1790 | 1828 | 1953 | 1730 | 1666 | 1756 | 1743 | 935 | 1325 | 683 | 1394 | 1449 | 1399 | 1215 | 1857 | 1671 | 2028 | 2065 | 2883 | 2910 |
| ANKRD1 | 1385 | 1894 | 1142 | 901 | 100 | 118 | 589 | 4163 | 6866 | 8021 | 9142 | 15829 | 17477 | 16404 | 389 | 375 | 886 | 524 | 151 | 2175 | 4415 | 3995 | 2833 | 1470 |
| INHA | 0 | 2 | 7 | 2 | 14 | 41 | 18 | 12 | 46 | 15 | 19 | 8 | 15 | 26 | 0 | 0 | 0 | 0 | 0 | 0 | 17 | 20 | 17 | 19 |
| MST1 | 24 | 17 | 32 | 31 | 307 | 295 | 391 | 327 | 181 | 684 | 669 | 1253 | 895 | 1258 | 18 | 27 | 26 | 44 | 25 | 18 | 14 | 27 | 33 | 38 |
| YAP1 | 3886 | 3734 | 2817 | 3422 | 2686 | 2606 | 2180 | 2742 | 3082 | 2589 | 2641 | 2242 | 2623 | 2439 | 1421 | 1381 | 1699 | 1346 | 1620 | 1894 | 2438 | 2497 | 2597 | 2688 |
| NF2 | 3317 | 3486 | 1854 | 3005 | 1260 | 1432 | 1297 | 1263 | 1417 | 766 | 758 | 1010 | 1035 | 1073 | 510 | 538 | 485 | 358 | 433 | 485 | 2033 | 2005 | 2932 | 2846 |
| LATS1 | 736 | 672 | 651 | 503 | 1803 | 1629 | 1027 | 1323 | 2002 | 1064 | 1065 | 862 | 1022 | 839 | 626 | 616 | 648 | 598 | 615 | 751 | 919 | 872 | 921 | 974 |
| LATS2 | 432 | 501 | 518 | 450 | 311 | 311 | 399 | 556 | 533 | 784 | 840 | 806 | 800 | 992 | 338 | 333 | 321 | 370 | 263 | 477 | 375 | 427 | 396 | 354 |
| SAV1 | 402 | 400 | 498 | 399 | 260 | 209 | 253 | 263 | 284 | 297 | 263 | 184 | 207 | 193 | 372 | 405 | 336 | 401 | 533 | 319 | 303 | 361 | 411 | 344 |
| TAZ | 356 | 420 | 406 | 318 | 224 | 218 | 232 | 225 | 190 | 164 | 160 | 110 | 129 | 157 | 459 | 427 | 493 | 423 | 731 | 477 | 337 | 367 | 496 | 467 |
| MOB1A | 2829 | 2835 | 3550 | 3540 | 3563 | 3213 | 2528 | 3289 | 3514 | 1611 | 1623 | 1322 | 1326 | 1548 | 2055 | 2014 | 2546 | 2026 | 2229 | 2314 | 3099 | 3284 | 3039 | 2848 |
| MOB1B | 448 | 467 | 406 | 610 | 328 | 315 | 305 | 269 | 478 | 1111 | 1233 | 1403 | 1082 | 1189 | 353 | 367 | 331 | 359 | 286 | 313 | 456 | 593 | 494 | 687 |

Legend (color coding):
- <0.5
- 0.5–0.66
- 0.66–1.5
- 1.5–2
- \>2

- **YAP/TAZ** target genes
- **Hippo** signaling components

**C**

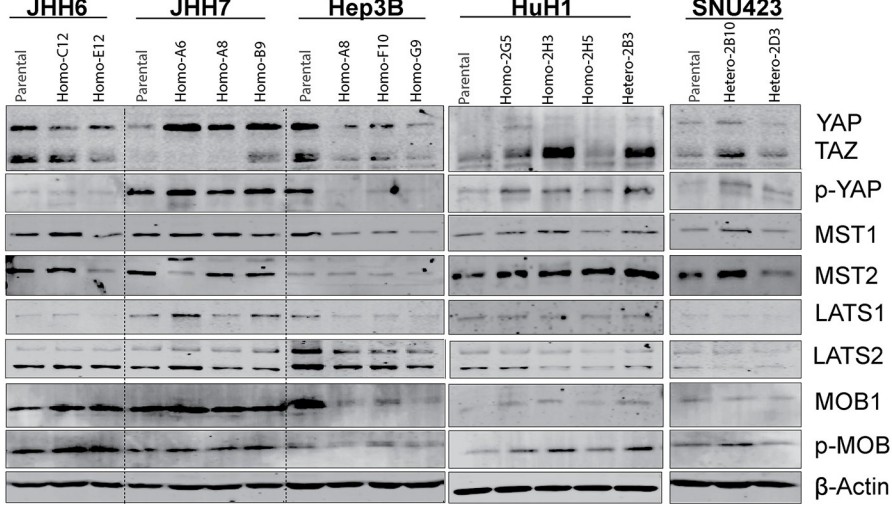

**Fig 6. Detailed expression analysis of β-catenin and YAP/TAZ signaling pathways.** (A) A 23-gene signature as reported by Abitbol et al. [12], was utilized to explore β-catenin signaling in liver cancers, including canonical, liver-specific, and negatively regulated β-catenin target genes. The red/green color coding for the repaired clones, was created by determining the fold-change in expression

relative to the corresponding average of parental samples. This color coding is only depicted for exploratory purposes and does not indicate that expression levels are also significantly different from the parental cells. (B) Similarly, the gene expression level was analyzed using RNA-seq data of YAP/TAZ target genes and components of the Hippo signaling pathway. (C) Western blotting for Hippo pathway components, with β-actin used as loading control. All protein density levels are normalized to β-actin in the same blot, and are presented in S10 Fig.

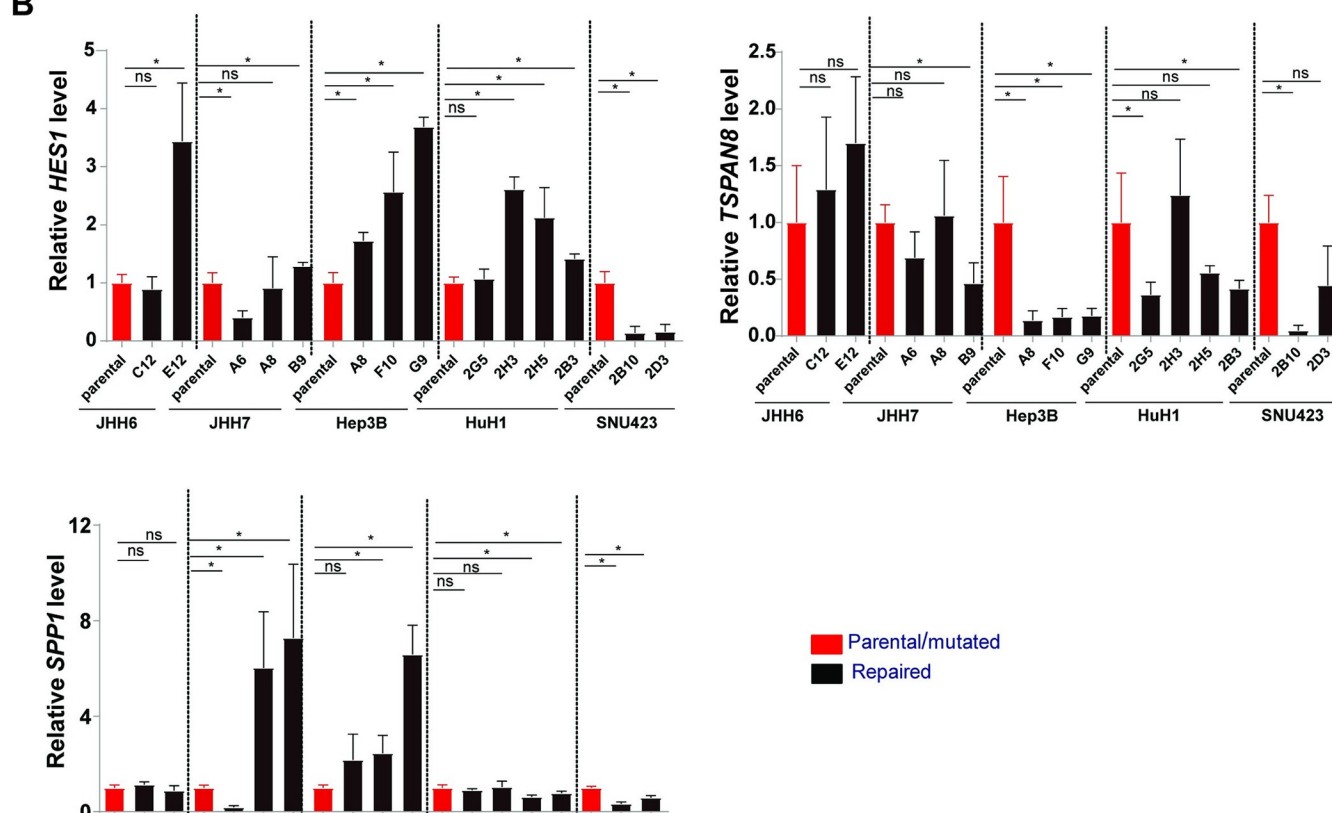

**A**

| GeneName | JHH6 | | | | JHH7 | | | | Hep3B | | | | HuH1 | | | | | SNU423 | | | |
|---|---|---|---|---|---|---|---|---|---|---|---|---|---|---|---|---|---|---|---|---|---|
| | parental | C12 | E12 | | parental | A6 | A8 | B9 | parental | A8 | F10 | G9 | parental | 2B3 | 2G5 | 2H3 | 2H5 | parental | 2B10 | 2D3 | |
| CD63 | 4854 | 4580 | 5627 | 4936 | 4604 | 4613 | 5259 | 5227 | 5004 | 4906 | 4663 | 6902 | 4785 | 6674 | 4883 | 4643 | 5884 | 6800 | 7261 | 5954 | 8679 | 8800 | 10413 | 11885 |
| SPP1 | 3156 | 2785 | 1141 | 882 | 38 | 48 | 3 | 2 | 170 | 146 | 166 | 64 | 76 | 284 | 45138 | 45128 | 53959 | 34892 | 39601 | 31375 | 1029 | 1053 | 742 | 730 |
| CDKN3 | 1795 | 1498 | 1300 | 2717 | 1025 | 539 | 815 | 664 | 437 | 460 | 418 | 274 | 389 | 323 | 1155 | 1042 | 1140 | 1133 | 1324 | 789 | 1293 | 985 | 1448 | 1119 |
| TSPAN8 | 15 | 8 | 5 | 21 | 1520 | 1336 | 2241 | 2611 | 901 | 173 | 179 | 95 | 53 | 177 | 1450 | 1479 | 1661 | 1300 | 2191 | 1374 | 0 | 0 | 0 | 0 |
| SOX9 | 153 | 108 | 143 | 320 | 2670 | 2857 | 2215 | 3209 | 1559 | 2504 | 2858 | 2139 | 2917 | 2474 | 130 | 154 | 368 | 268 | 219 | 761 | 1864 | 1860 | 1905 | 1852 |
| HEY1 | 54 | 53 | 61 | 44 | 227 | 194 | 95 | 52 | 251 | 41 | 45 | 13 | 28 | 26 | 25 | 35 | 13 | 14 | 4 | 19 | 3 | 11 | 11 | 30 |
| HES1 | 103 | 91 | 48 | 94 | 517 | 612 | 498 | 614 | 967 | 586 | 647 | 589 | 558 | 567 | 411 | 480 | 435 | 346 | 651 | 376 | 24 | 29 | 10 | 19 |
| HEYL | 0 | 0 | 0 | 1 | 23 | 21 | 19 | 12 | 39 | 65 | 63 | 74 | 96 | 99 | 22 | 7 | 10 | 5 | 0 | 14 | 0 | 0 | 0 | 0 |
| NOTCH1 | 391 | 395 | 291 | 463 | 260 | 279 | 49 | 42 | 262 | 373 | 262 | 141 | 249 | 207 | 681 | 653 | 1010 | 968 | 1441 | 1120 | 312 | 318 | 190 | 185 |

| | |
|---|---|
| ■ | <0.5 |
| ■ | 0.5-0.66 |
| ■ | 0.66-1.5 |
| ■ | 1.5-2 |
| ■ | >2 |

**Fig 7. Detailed expression analysis of the Notch signaling pathway.** (A) Expression level of Notch target genes was analyzed using RNA-seq data. Color coding was identical as described for Fig 6. (B) QRT-PCR assay shows the relative mRNA expression levels of *HES1*, *TSPAN8*, and *SPP1*. The data was normalized to the housekeeping gene *GAPDH* (n = 2, two independent experiments). Additionally, the data was further normalized to the corresponding parental cell line, with the parental expression set to 1. Statistical significance for all experiments was analyzed using a Mann-Whitney test (*$P<0.05$).

YAP/TAZ and Notch signaling pathways and low/absent activation of β-catenin signaling [11, 12]. However, given the long-term process of tumor development, such tumors will have acquired many additional (epi)genetic alterations that cooperate with AXIN1 mutation for successful tumor formation. This will obscure the direct consequences of AXIN1 mutation to support tumor growth. Therefore, we decided to investigate AXIN1-mutant HCC cell lines in which we restored endogenous AXIN1 expression through gene editing. This allowed us a direct side-by-side comparison of cellular features affected by AXIN1 mutation. We used five cell lines representative of mesenchymal- and hepatoblast-like subtypes [28].

Moreover, we chose to use AXIN1-mutant HCC cell lines instead of organoid models for our study due to limitations with organoids in liver cancer research. First, organoids have low success rates in establishing long-term models for HCC, and to the best of our knowledge, currently no organoid models exists that are derived from an AXIN1-mutant HCC [33–35]. Second, it remains difficult to genetically modify them when the desired alteration does not provide a selective growth advantage. Third, most organoids require a continuous exposure to reagents, such as Wnt and R-Spondin ligands, that maintain sufficient β-catenin signaling for supporting their growth. This may obscure the requirement for low level β-catenin signaling imposed by AXIN1 mutations that we wish to investigate. HCC cell lines, on the other hand, have been found to closely resemble aggressive forms of liver cancer and have been used effectively to test anticancer agents and identify therapeutic response markers [28]. Therefore, using AXIN1-mutant HCC cell lines is a practical and relevant approach for studying the impact of these mutations.

Overall, our analysis revealed the following: (1) in all five cell lines, restoring endogenous AXIN1 expression leads to reduced in vitro growth characteristics, confirming its role as a genuine tumor suppressor for liver cancer; (2) in all cases this is accompanied by reduced β-catenin signaling; (3) however, increasing β-catenin signaling in AXIN1-repaired clones to levels comparable with the AXIN1-mutant parental cells, does not or only partially revive growth in the repaired clones. This indicates that AXIN1-associated alterations in β-catenin signaling are not solely responsible for the tumor promoting effects of AXIN1 mutation; (4) RNA sequencing does not indicate a specific set of genes/pathways that are consistently affected by AXIN1 mutation.

For all five HCC cell lines we successfully obtained independent clones with restored AXIN1 expression. This shows that the original AXIN1 mutation is not absolutely essential to sustain growth in culture. Such an observation is not unprecedented as this has been more often observed for mutated oncogenes or tumor suppressor genes. For example, complete inhibition of the oncogenic BRAF[V600E] mutation in colorectal cancers leads to upregulation of EGFR expression, which compensates for loss of mutant BRAF [36, 37]. Likewise, constitutive β-catenin activation was considered essential for colorectal cancers, but several reports have shown that these cancers are less dependent on continued β-catenin signaling for their growth than generally believed [38–40]. Also, in our case the AXIN1 mutation is not absolutely required for growth of five liver cancer cell lines, but the reduced growth characteristics that we observe, show that its loss contributes to cancer cell fitness. However, our results also indicate that no straightforward explanation can be provided through which AXIN1 mutation supports tumor growth.

Originally, given its role in the β-catenin destruction complex, AXIN1 mutation was mainly considered to contribute to tumorigenesis by activating β-catenin signaling. This view was challenged by reports suggesting that AXIN1 mutation leads to liver cancer without clearly activating β-catenin signaling [10–12]. However, our current results and those of other publications clearly show that inactivating AXIN1 leads to increased β-catenin signaling, albeit moderate [12, 13, 17]. A β-catenin reporter construct, which is almost exclusively dependent

on nuclear β-catenin for activation, was reduced in activity in all clones with AXIN1 repair. Likewise, we observed reduced expression of more responsive canonical β-catenin target genes, such as *AXIN2*, *NOTUM*, and *NKD1*. Other canonical target genes show a less consistent response, and this was even more the case for "so-called" liver-specific β-catenin target genes.

In contrast to the reporter construct, all endogenous target genes will be co-regulated by various additional transcription factors besides TCF/β-catenin. As such, they are expected to require higher levels of nuclear β-catenin to significantly affect their expression level. Because of partial functional redundancy with AXIN2 in the breakdown complex, AXIN1 mutation is expected to result in a modest nuclear β-catenin signaling and alteration of gene expression, which is what we observe. Whether these changes are sufficient to promote tumorigenesis is difficult to conclude with certainty, but our current results are partially in line with the Paris team who propose that AXIN1 deficiency does not support HCC growth solely by activating β-catenin signaling [10, 12]. It is also in line with a report by Ding et al. who showed that knockdown of β-catenin reduced the growth of Hep3B cells, but not that of JHH6, JHH7 and HuH1, indicating that β-catenin is not equally relevant for all AXIN1-mutant HCC cell lines [41]. The growth characteristics of most AXIN1-repaired clones could not or only partially be improved by increasing β-catenin signaling through Wnt-addition. Possibly, merely adding access Wnt3A ligand in the culture medium does not entirely recapitulate the alteration in endogenous β-catenin signaling induced by AXIN1 mutation, but given the minor changes in β-catenin target gene expression, it seems more likely that AXIN1 mutation contributes to liver cancer through other mechanisms than solely activating β-catenin signaling.

One such mechanism has been proposed to be YAP/TAZ signaling. Increased activation of this pathway is observed in more than 60% of HCCs, so is not exclusively associated with AXIN1 mutation [42]. However, AXIN1 mutant HCCs appear to be more commonly associated with increased YAP/TAZ signaling than other HCC subtypes [12]. In addition, they seem more strongly dependent on concomitant activation of this pathway to sustain tumor growth. Among others this was shown in cMET/AXIN1-KO-driven HCC mouse models, which were strongly blocked in their growth by simultaneous inactivation of YAP/TAZ, while this was less the case for cMET/β-catenin induced cancers [32]. Both tumor groups showed a near-complete loss of LATS1/2 expression, critical kinases for YAP/TAZ turnover, which will have a strong impact on YAP/TAZ signaling. Nevertheless, the weaker tumorigenic properties of AXIN1 mutation may lead to the selection of additional mechanisms that more strongly activate YAP/TAZ signaling than required in β-catenin mutant cancers. One such mechanism was proposed to be the AXIN1 mutation itself. AXIN1 can bind to YAP/TAZ at its C-terminal half, thereby modulating β-catenin breakdown [30, 31]. Liang et al. suggested however that through this association, AXIN1 also leads to reduced levels of YAP/TAZ proteins [32]. To this aim, they used short-term experiments in which AXIN1 levels were modulated by siRNA mediated knockdown or overexpression, which indeed suggested that AXIN1 is involved in YAP/TAZ turnover. However, in our AXIN1-repaired clones of 5 different cell lines we do not observe such a correlation. None of the repaired clones shows a consistent decrease in expression of YAP/TAZ target genes. In addition, YAP/TAZ levels are unchanged or even increased in the AXIN1-repaired clones. The only exception appears to be Hep3B in which YAP/TAZ levels are decreased following AXIN1-repair, but this is surprisingly accompanied with increased expression of target genes. A possible explanation for this apparent contradiction may reside in the use of established clones vs short-term experiments, that is, prolonged culture may select for mechanisms that re-activate YAP/TAZ signaling in AXIN1-repaired clones. Alternatively, AXIN1 mutation does only lead to increased YAP/TAZ levels in a subset of AXIN1-mutant HCCs. YAP/TAZ signaling can be regulated at multiple levels [30], and it seems likely that

differences exist between individual HCC tumors and cell lines in the manner they activate YAP/TAZ signaling.

Except for a moderate activation of β-catenin signaling, our RNA-seq and pathway analysis fails to identify a straightforward explanation for the tumorigenic properties of AXIN1 mutation. No other gene or pathway is consistently affected by the AXIN1 mutation status. Possibly this weak β-catenin activation requires cooperation with other genes/pathways that are more or less unique to each cell line that we studied. HCCs do not follow one specific route to malignancy, meaning that each cancer carries a defined set of (epi)genetic alterations operating in concerted action with the AXIN1 mutation. In support of this, principal component analysis shows that cell line identity is more dominant in determining the RNA expression profile than AXIN1 mutation status. A second possibility may reside in functions of AXIN1 not directly related to regulating cellular signaling. For example, AXIN1 has also been implicated in centrosomal biology in a handful of papers [43–45]. AXIN1 was shown to co-localize with centrosomes through associating with γ-tubulin, and its knockdown resulted in reduced microtubule nucleation from the centrosome [43]. Also in Drosophila embryos, AXIN appears to contribute to mitotic fidelity [44]. Thus, by restoring AXIN1 expression one could speculate that mitotic checkpoints are more robustly activated during mitosis, thereby delaying mitosis and effectively resulting in longer cell cycle times and overall reduced growth.

In conclusion, our results show that restoring endogenous AXIN1 expression in five AXIN1-mutant HCC cell lines, in all cases leads to reduced growth characteristics and reduced β-catenin signaling. However, we cannot formally prove that the latter is also responsible for the reduced growth. Exploring other genes or pathways (e.g., YAP/TAZ, Notch) potentially affected by AXIN1 mutation through RNA-sequencing and in detail analysis of candidate pathways, did not reveal a consistent explanation linked to AXIN1. In fact, the recently proposed link between AXIN1 and direct regulation of YAP/TAZ proteins and signaling, could not be confirmed in our five cell lines. This can either be explained by differences in methodology (short term alterations of AXIN1 levels vs long term establishment of clones) or indicates that it is restricted to a subset of HCC cell lines and cancers. The latter may be more generally true for AXIN1 mutation, i.e., its mechanism of action could differ depending on (epi)genetic alterations specific for each tumor. Whichever mechanism(s) may be uncovered in the future, it may possibly cooperate with the moderate activation of β-catenin signaling that we consistently observe in AXIN1-mutant HCC cell lines.

## Supporting information

**S1 Fig.** A) Baseline information of HCC cell lines used in this study, including differentiation subtype as reported by Caruso et al. [28], puromycin concentration used to select clones, and the type of AXIN1 mutation. Microscopic images are obtained from the Zucman lab website (https://lccl.zucmanlab.com/hcc/cellLines). (B) Original sequence chromatograms depicting the mutations observed in each cell line.
(PDF)

**S2 Fig. An AXIN1 expression construct was generated that carries the same D94_Q108 deletion as present in the JHH7 cell line.** (A) When expressed in HEK293T cells it leads to a strong increase in β-catenin reporter activity comparable to adding exogenous Wnt3a ligand, while wild-type AXIN1 has no effect. This shows that the D94_Q108del variant protein interferes with β-catenin regulation in a dominant manner when overexpressed. (B) A co-immunoprecipitation experiment with GFP-tagged APC (aa 1199–2167) shows that the D94_Q108del variant is unable to bind APC. Statistical significance was analyzed using a Mann-Whitney test (****$P < 0.0001$). These data have also been described elsewhere (ref 18), but are reproduced

here for clarity.
(PDF)

**S3 Fig. Sequence chromatograms from all successfully repaired clones.**
(PDF)

**S4 Fig. Expression levels of NOTUM mRNA in AXIN1-repaired clones.** qPCR was used to measure NOTUM expression levels. With the exception of JHH6-repaired-E12, the expression level of NOTUM was found to be lower in the AXIN1-repaired clones compared to the parental cells. The data were normalized to the housekeeping gene GAPDH, with the parental values set to 1. The statistical significance of the results was analyzed using the Mann-Whitney test, with the level of significance indicated as follows: (*P < 0.05, **P < 0.01, ***P < 0.001, ***P < 0.0001).
(PDF)

**S5 Fig. The impact of siAXIN2 on the β-catenin signaling was evaluated in both parental and repaired clones of the HCC cell line.** To this aim, a β-catenin reporter assay was performed. The β-catenin reporter activities are presented as WRE/CMV-Renilla ratios (mean ± SD, n = 3, two independent experiments). The values depicted here, were used to determine the siAXIN2/siControl ratios shown in Fig 2D. All values were scaled to log10.
(PDF)

**S6 Fig. Representative images of the colony formation assay performed on all parental and AXIN1-repaired clones.**
(PDF)

**S7 Fig. RNA-seq analysis for AXIN1-repaired clones and their corresponding parental HCC cell lines.** (A) The RNA sequencing data were subjected to principal component analysis, which clearly separated the samples into five distinct groups based on cell line identity. (B) Volcano plots for each cell line separately, showing genes significantly altered at least log2 fold change of 1 in expression (p <0.01).
(PDF)

**S8 Fig. A KEGG analysis does not reveal a pathway consistently altered in all cell lines.**
(PDF)

**S9 Fig. QRT-PCR assay shows the relative HMMR mRNA expression levels.** The data was normalized to the housekeeping gene GAPDH (mean ± SD, n = 2, two independent experiments). Additionally, the data was further normalized to the corresponding parental cell line, with the parental expression set to 1. Statistical significance for all experiments was analyzed using a Mann-Whitney test (*P < 0.05).
(PDF)

**S10 Fig. Quantification of band intensities for indicated proteins for the immunoblots shown in Fig 6C.**
(PDF)

**S1 Raw images. Original immunoblots.**
(PDF)

**S1 File.**
(XLSX)

**S2 File.**
(XLSX)

**S3 File.**
(XLSX)

**S4 File.**
(XLSX)

**S1 Table. Primer sequences for amplifying genomic AXIN1 sequences and cloning into TA vector.**
(PDF)

**S2 Table. AXIN1 sgRNAs info.**
(PDF)

**S3 Table. Primers to introduce silent and PAM-site mutations in repair HDR plasmids.**
(PDF)

**S4 Table. Primers to identify correctly repaired AXIN1 in cell clones.**
(PDF)

**S5 Table. Primer sequences for amplifying AXIN1 cDNA.**
(PDF)

**S6 Table. Primer sequences used for Qpcr.**
(PDF)

**S7 Table. Genes significantly up- and downregulated in AXIN1-repaired HCC cell lines.**
(XLSX)

## Acknowledgments

We wish to thank Dr. Sandra Rebouissou, Paris, France, for generously providing the JHH6, JHH7, HuH1, and SNU423 cell lines.

## Author Contributions

**Data curation:** Ruyi Zhang, Shanshan Li, Kelly Schippers, Boaz Eimers, Ron Smits.

**Formal analysis:** Ruyi Zhang, Bastian V. H. Hornung, Mirjam C. G. N. van den Hout, Wilfred F. J. van Ijcken.

**Methodology:** Jiahui Niu.

**Resources:** Ruyi Zhang.

**Software:** Ruyi Zhang, Wilfred F. J. van Ijcken.

**Supervision:** Maikel P. Peppelenbosch, Ron Smits.

**Writing – original draft:** Ruyi Zhang.

**Writing – review & editing:** Ruyi Zhang, Ron Smits.

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
