## [Decision Letter · Decision Letter 0]

18 Mar 2024

PONE-D-23-31480Unraveling the Impact of AXIN1 Mutations on HCC Development: Insights from CRISPR/Cas9 repaired AXIN1-mutant liver cancer cell linesPLOS ONE

Dear Dr. Smits,

Thank you for submitting your manuscript to PLOS ONE. After careful consideration, we feel that it has merit but does not fully meet PLOS ONE’s publication criteria as it currently stands. Therefore, we invite you to submit a revised version of the manuscript that addresses the points raised during the review process.

We look forward to receiving your revised manuscript.

Kind regards,

Michael Klymkowsky, Ph.D.

Academic Editor

PLOS ONE

Journal Requirements:

Did you know that depositing data in a repository is associated with up to a 25% citation advantage (https://doi.org/10.1371/journal.pone.0230416)? If you’ve not already done so, consider depositing your raw data in a repository to ensure your work is read, appreciated and cited by the largest possible audience. You’ll also earn an Accessible Data icon on your published paper if you deposit your data in any participating repository (https://plos.org/open-science/open-data/#accessible-data).

"This research was financially supported by a China Scholarship Council PhD fellowship to Ruyi Zhang (File NO. 201808530490), Shanshan Li (File NO. 201909370083) and Jiahui Niu (File NO. 202007660001)."

5. Please note that funding information should not appear in the Acknowledgments section or other areas of your manuscript. We will only publish funding information present in the Funding Statement section of the online submission form. Please remove any funding-related text from the manuscript. 

6. Please note that your Data Availability Statement is currently missing the direct link to access each database. If your manuscript is accepted for publication, you will be asked to provide these details on a very short timeline. We therefore suggest that you provide this information now, though we will not hold up the peer review process if you are unable.

7. In this instance it seems there may be acceptable restrictions in place that prevent the public sharing of your minimal data. However, in line with our goal of ensuring long-term data availability to all interested researchers, PLOS’ Data Policy states that authors cannot be the sole named individuals responsible for ensuring data access (http://journals.plos.org/plosone/s/data-availability#loc-acceptable-data-sharing-methods).

**Additional Editor Comments:**

The reviewers have many a number of specific comments and a few suggestions.  Each comment (and call for more detail and consistency) should be explicitly addressed in an accompanying "Response to reviewer" letter.  

Reviewers' comments:

Reviewer's Responses to Questions

**Comments to the Author**

1. Is the manuscript technically sound, and do the data support the conclusions?

Reviewer #1: Partly

Reviewer #2: Partly

2. Has the statistical analysis been performed appropriately and rigorously? 

Reviewer #1: Yes

Reviewer #2: I Don't Know

3. Have the authors made all data underlying the findings in their manuscript fully available?

Reviewer #1: Yes

Reviewer #2: No

4. Is the manuscript presented in an intelligible fashion and written in standard English?

Reviewer #1: Yes

Reviewer #2: Yes

5. Review Comments to the Author

Reviewer #1: In the manuscript the authors argue that they have successfully repaired AXIN1 utilising CRISPR/Cas9 mediated HDR in 5 HCC cell lines (JHH6, JHH7, Hep3B, HuH1, and SNU423). They show AXIN1 repair reduces b-catenin signalling, cell viability and colony formation. Their data suggest that Wnt3a stimulation of the repaired clones restored b-catenin activity and, in some lines colony numbers, but not overall cell growth. RNAseq data revealed cell-line specific responses to AXIN1 repair overlap were restricted to a small number of classic Wnt targets including AXIN2. Changes to AXIN1 did not appear to correlate with alterations to the Yap/Taz and Notch pathways despite prior data suggesting their involvement.

Although the data looks interesting, the results were somewhat frustrating to follow in their current state due to the lack of clarity in the methods describing how Wnt pathway activity was / was not maintained during the development and subsequent analysis of the clones being studied.

Major issue

The level of Wnt pathway activity was the key focus for the manuscript based on the hypothesis that restoration of Axin activity would downregulate TCF-dependent transcription. If the Wnt pathway were to behave as a digital ON/OFF switch, one might expect that the restoration of AXIN1 function would induce differentiation and perhaps prevent the emergence of clones. This possibility was mitigated by culturing the clones during selection in a combination of Wnt and Rspo conditioned medium. However, it was unclear whether 10% Wnt3a and Rspo CM was used to maintain AXIN1 repaired clones or whether the repaired clones were not viable without the CM. The media conditions for the subsequent experiments were not made clear. Furthermore, Methods (Line 204) describes the use of RPSO CM alone and in combination with WNT3a CM however the results are only shown for Wnt3a CM– does addition of RSPO CM restore growth? A much more detailed discussion, exploration and explanation of the possible concerns / interpretation is required in addition to provision of the missing technical information.

Suggestions to improve strength of conclusions

Although the observations (assuming the point above is adequately addressed) are interesting, they have to be interpreted with great caution due to the possible concerns covered here:

If the Wnt pathway were behaving as a variable-level rheostat (Cf. ON/OFF), there may be strong selective pressure for genetic or epigenetic modulation of Wnt pathway activity to maintain a ‘just right’ level for each individual cell type. Given that this interpretation is also consistent with the data, it would have been good to have seen the use of cell clones derived from a similar transfection / selection process to ensure that the selection methods used did not lead to variation in Wnt pathway activity in processes that were unlinked to the restoration of AXIN1 activity. Related to this point, studies could also have been done to show that reducing AXIN1 levels in the clones (e.g. using shRNA or siRNA) reversed the effects of the rescue since this would exclude the ‘clonal variation’ explanation.

It would also have been useful to have seen some studies carried out using purified Wnts / Rspo proteins and efforts to monitor levels in the culture media since some of the variable responses described between cell lines might be related to the rates at which active Wnt/Rspo-ligands are maintained / depleted in culture.

Minor points:

How was the AXIN1 antibody determined to be against C-terminus of the protein

Line 277-278 How homozygosity or heterozygosity of repair was not explained. And no sequencing data presented to show the repair of AXIN1.

Figure 2 B, C, D cell line labels missing above most of the charts.

Figure 4 A β-catenin signalling activity missing statistics.

Figure 4 B is missing error bars

Figure 6 C. Quantification of protein levels required. Additionally, keep order of cell lines same as in previous figures (SNU423 and HuH1 blots switched)

Methods describe assays not shown in the results:

Statistical analysis – methods state two tailed students t test used to compare between two groups while non parametric test – Mann-Whitney U test used for groups with non-normally distributed variables. Figure legends indicate T Test not used once. Not clear what data points are being shown in charts – is it n in each experiment or the mean from each experiment.

Line 378-379 KEGG analysis – data not shown?

Line 164-167 – Methods indicate use of fluorescent western blotting – no fluorescent blots shown in results

Sup Table G qPCR – primers for Axin1 exons 7-8. No AXIN1 expression data shown. Also SERPINE1 primers but no data

Reviewer #2: In the manuscript “Unraveling the Impact of AXIN1 Mutations on HCC Development: Insights from CRISPR/Cas9 repaired AXIN1-mutant liver cancer cell lines”, Zhang et al examined phenotypes of 5 human HCC cell lines with restored wild-type AXIN1. The study examined 2-4 clones for each cell line and found reduced beta-catenin activity, cell viability, and colony formation. Restored beta-catenin activity with Wnt stimulation did not overcome growth defects, suggesting that activities other than beta-catenin are involved in this phenotype. Canonical Wnt target genes were not consistently altered in each of the rescued clones, again consistent with effects of AXIN1 mutations beyond the regulation of beta-catenin signaling. Further, RNAseq and western blot analysis did not support a consistent pathway regulated by AXIN1 in each line, including YAP/TAZ or Notch.

Though the manuscript is clearly written and logical, some experiments were not conducted rigorously, with appropriate replication, and sample sizes. For Fig. 2D, Fig. 3B, Fig. 4 A and B, Fig. 5, and Fig. 7B, the experiments were only performed two independent times. Performing three times is required to increase rigor.

The authors have made some of the data supporting the findings in their manuscript available. However, other data should be expanded to help readers more fully understand and interpret the unexpected results with the cell phenotypes. Notably, the experimental details for how the AXIN1 mutations in the HCC lines were identified and confirmed are lacking. Some of the mutations differ from those described in publicly available databases such as “the Liver Cancer Cell Line Database”. The precise mutations as well as their hetero- or homozygosity become important when considering potential explanations for the unexpected results. Another dataset that would increase the rigor of the study is to show the complete length of the western blot of the parental cell line lysates probed with both a C-terminal and an N-terminal AXIN1 antibody (Fig. 2C). This would support the statement on line 255 that the mutations lead to short, truncated proteins. Likewise, it would be good to show more of the gel and molecular weight markers to confirm the size of the restored Axin1 product in Fig. 2A. If these blots were probed with an N-terminal AXIN1 antibody it could be demonstrated that truncated AXIN1 was eliminated in all but SNU423 cells (as stated in the text). The level of Axin1 restored appears to be quite variable- eg. HuH1 and SNU423 are quite faint. Are the restored Axin1 levels comparable to endogenous levels in HEK203T, or other HCC lines with wild-type Axin1? Fig. 2D has a label “JHH7” in the middle of the panel. Why is the GFP-APC band shifted in supplementary fig 2B depending on which FLAG-AXIN1 it was immunoprecipitated with?

Fig. 1D Are WRE and MRE both controlled for transfection (CMV-renilla) when presented as WRE/MRE? The level of AXIN2 mRNA is not significantly reduced in the AXIN1-repaired clone JHH6-E12 (in contrast to line 282 statement) which also showed no decrease in NOTUM mRNA level (sup Fig. 3). How is the large (1000x) range of results explained in Fig. 2D? In other words, why is there a ~5-fold difference in beta-catenin reporter activity with AXIN2 knockdown in parental Hep3B cells and a ~5000-fold difference in parental HuH1 cells? This could be considered in the discussion. The statement in line 292 seems to miss the mark—the AXIN1 mutant HCC cells appear more dependent on AXIN2 for beta-catenin regulation? For Fig. 2D, the values are shown “relative to the WRE/CMV-Renilla ratios obtained for the siControl-WT which is set to 1” However, there doesn’t appear to be anything shown as “1” in this figure or in Sup Fig 4. It is unclear how these values were obtained and what they mean. For sup Fig. 4, the X-axis is shown on a logarithmic scale, but the values themselves are not log10. The axis labeling is confusing on these graphs and those in Fig. 4A.

In Figure 3A and 4B, the growth rate is not being measured. With more time points between d3 and d7, the growth rate could be estimated and one could determine if 1) there is a lag until d3 and then all lines start to grow, but the parental grows more quickly or 2) there is a longer lag with the rescue clones and they are less able to grow in sparse conditions (consistent with the colony formation assay). Line 319- “suggest that mechanistically this is accomplished by increasing…”—no mechanism is shown, but there is a correlation with beta-catenin signaling. Line 347: “we cannot formally demonstrate that reduced beta-catenin signaling is also responsible for the observed change” seems misleading. Beta-catenin activity with added Wnt does not correlate with growth for every cell line. In some cases (eg. JHH6) adding Wnt results in fewer colony numbers.

For Fig. 6C, are these one blot for each cell-line set that was probed with 10 different antibodies? Were these blots stripped and re-probed? More experimental details in the methods section would be useful.

The experimental results are intriguing and unexpected. The authors should consider if the results in Fig. 5 are consistent with Ding et al, Oncotarget, 2017 “Oncogenic dependency on β-catenin in liver cancer cell lines correlates with pathway activation” who found that silencing beta-catenin led to growth defects in Hep3B cells but not in JHH6, JHH7 or HuH1 (supplementary Fig. 6, 10, 12, and 17). Can other properties of the 5 parental cells be considered to obtain more information from the data? For instance, 3 are from Japanese patients, 3 of the cell lines are associated with hepatitis virus infection, four have P53 mutations and 3 have TERT mutations.

Overall, this is an interesting study with intriguing results. The results are worth presenting in the most robust form possible, particularly since they contrast with other published results.

6. PLOS authors have the option to publish the peer review history of their article (what does this mean?). If published, this will include your full peer review and any attached files.

Reviewer #1: No

Reviewer #2: No

---

## [Author Response · Author response to Decision Letter 0]

26 Apr 2024

We have provided all requested information in the "Response to reviewers PONE-D-23-31480" file uploaded with this submission.

---

## [Decision Letter · Decision Letter 1]

15 May 2024

Unraveling the Impact of AXIN1 Mutations on HCC Development: Insights from CRISPR/Cas9 repaired AXIN1-mutant liver cancer cell lines

PONE-D-23-31480R1

Dear Dr. Smits,

We’re pleased to inform you that your manuscript has been judged scientifically suitable for publication and will be formally accepted for publication once it meets all outstanding technical requirements.

Kind regards,

Michael Klymkowsky, Ph.D.

Academic Editor

PLOS ONE

Additional Editor Comments (optional):

Reviewers' comments:

Reviewer's Responses to Questions

**Comments to the Author**

1. If the authors have adequately addressed your comments raised in a previous round of review and you feel that this manuscript is now acceptable for publication, you may indicate that here to bypass the “Comments to the Author” section, enter your conflict of interest statement in the “Confidential to Editor” section, and submit your "Accept" recommendation.

Reviewer #1: All comments have been addressed

Reviewer #2: All comments have been addressed

2. Is the manuscript technically sound, and do the data support the conclusions?

Reviewer #1: (No Response)

Reviewer #2: Yes

3. Has the statistical analysis been performed appropriately and rigorously? 

Reviewer #1: (No Response)

Reviewer #2: Yes

4. Have the authors made all data underlying the findings in their manuscript fully available?

Reviewer #1: (No Response)

Reviewer #2: Yes

5. Is the manuscript presented in an intelligible fashion and written in standard English?

Reviewer #1: (No Response)

Reviewer #2: Yes

6. Review Comments to the Author

Reviewer #1: (No Response)

Reviewer #2: The authors responded to my previous comments/concerns adequately. This paper should be published and the unexpected results considered when developing future models of AXIN1 mutation functions in HCC.

7. PLOS authors have the option to publish the peer review history of their article (what does this mean?). If published, this will include your full peer review and any attached files.

Reviewer #1: No

Reviewer #2: No

---

## [Editor Report · Acceptance letter]

29 May 2024

PONE-D-23-31480R1 

PLOS ONE

Dear Dr. Smits, 

I'm pleased to inform you that your manuscript has been deemed suitable for publication in PLOS ONE. Congratulations! Your manuscript is now being handed over to our production team.

Kind regards, 

on behalf of

Dr. Michael Klymkowsky 

Academic Editor

PLOS ONE